# Buckling suppression of a thin-walled Miura-origami patterned tube

Yangqing Liu [1,2,3]*, Jianguo Cai[3,4]*, Jian Feng[3,4]

**1** State Key Laboratory of Mountain Bridge and Tunnel Engineering, Chongqing Jiaotong University, Chongqing, China, **2** School of Civil Engineering, Chongqing Jiaotong University, Chongqing, China, **3** Key Laboratory of Concrete & Pre-stressed Concrete Structure of Ministry of Education, Southeast University, Nanjing, China, **4** School of Civil Engineering, Southeast University, Nanjing, China

* lyq19@cqjtu.edu.cn (YL); j.cai@seu.edu.cn (JC)

## Abstract

In order to improve energy absorption capacity of tubes under axial compression, this work introduces an octagon tube patterned by curved Miura origami pattern and aims at suppressing global buckling of the tube via an appropriate fold line scheme. By categorizing the fold lines of the tube into two types (hinge-like lines and continuous lines) and allocating them to different positions, four arrangement schemes of the lines are developed. Through numerical comparison in force-displacement curve, stress distribution and lateral deformation capacity among four 3-level patterned tubes under different schemes, Scheme 4 (inclined valley lines as hinge-like lines and the others as continuous lines) is found to outperform the others in suppressing global buckling by reducing the magnitude of lateral deformation by up to 59.9% and by delaying the occurrence of global buckling by up to 35.9% compared with the other schemes. To step further, the scheme is applied in a long tube and geometrically different tubes are compared. The results prove that the scheme is potentially an effective way to alleviate buckling instability of a long tube when appropriately designed.

## Introduction

Origami is an ancient art of forming planar materials to spatial structures by folding the materials along predetermined fold lines. The intricate structures gain unconventional properties and yield origami-inspired engineering applications including soft robotics [1, 2], self-folding structures [3, 4], biomechanical materials [5, 6] and so on. Among the applications, energy absorbers aim at precluding global buckling and achieving yielding along the entire length by introducing origami patterns as geometrical imperfections.

The concept of origami-inspired energy dissipaters is built on the truth that thin-walled tubes are widely accepted as energy dissipaters for their excellent mechanical performance, manufacturing ease and affordability. Straight square tubes and circular tubes are the most common types of energy dissipaters. Large amount of efforts has been made to investigate the mechanical behaviors of the two types of tubes under axial crushing. Through experimental and numerical study, the deformation modes and energy dissipation characteristics have been investigated and analytical theories have been accordingly proposed [7–9]. Also, tubes with other cross sections have been explored, such as more polygonal tubes [10] and cone tubes

**Data Availability Statement:** All relevant data are within the article and its Supporting Information files.

**Funding:** This research was funded by: 1. The Natural Science Foundation of China, grant number 52008064, YL. 2. The Natural Science Foundation

of China, grant number 51822805, JC. 3, The Science and Technology Research Program of Chongqing Municipal Education Commission of China, grant number KJQN202000737, YL. 4, Open Project of Key Laboratory of Concrete and Prestressed Concrete Structure of Ministry of Education of China, grant number CPCSME2019-07, YL. The funders had no role in study design, data collection and analysis, decision to publish, or preparation of the manuscript.

**Competing interests:** The authors have declared that no competing interests exist.

[11] and star-shaped tubes [12]. These tubes show highly sensitivity to initial geometrical imperfections and the unpredictable deformation deteriorates the performance under loading. However, at the same time, the study reveals that appropriate imperfections trigger specific deformation modes and consequently alter the response under loading. That inspires the introduction of origami pat-terns into thin-walled tubes. Wu and Hagiwara [13] applied Kresl-ing pattern on aluminum tubes to absorb impact shock and an optimization method of the energy absorption capacity was put forward. Song and Chen [14] proposed a type of tubes with equilateral trapezoid patterns, which were numerically and experimentally verified to improve the crashworthiness by successfully inducing the crushing modes of the tubes. Since then, various types of origami patterns have been designed and applied to tubes, including Miura origami pattern [15, 16], Yoshimura pattern [17], crash box [18, 19], kite-shape pattern [20, 21], waterbomb pattern [22, 23] and so on. These elaborately designed patterns signifi-cantly enhance the energy absorption capacity of the tubes. The above-mentioned work mean-ingfully contributes to the understanding of the applications of origami-based tubes. However, the work is limited to short tubes, for which global buckling is not a problem. When it comes to longer tubes, global buckling is more likely to occur and in turn adversely affects the behav-ior in terms of significant compression capacity loss, strain concentration at mid-span, sudden force redistribution and material fracture [24]. Therefore, it is necessary to extend the current work to longer tubes.

In this work, an octagon tube patterned by curved Miura origami is introduced and an arrangement scheme of the fold lines is proposed to suppress global buckling of the tube. Through numerical analysis, the superior scheme is proved to be potentially an effective way to alleviate buckling instability of a long tube when appropriately designed.

The main structure of the manuscript is as follows: In the 2nd section, a curved Miura ori-gami-patterned unit is introduced. In the 3rd section, four arrangement schemes of fold lines are developed and the superior one is picked by comparisons of their effects on buckling sup-pression of the tubes. In the following section, application of the superior scheme to a long tube is given. Finally, conclusions drawn in this work are summarized.

## Materials and methods

### Curved Miura origami-patterned unit

Curved Miura origami, a.k.a. two-fold Miura origami, is a variant of a regular Miura origami. The difference between a curved Miura origami and a regular Miura origami lies in the angles $\varphi_1$ and $\varphi_2$, as shown in Fig 1, where the solid lines represent mountain fold lines, the dash lines

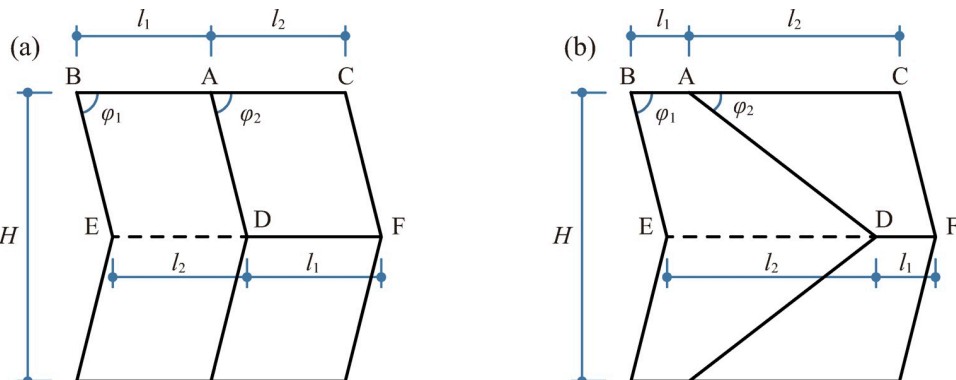

**Fig 1. A regular Miura origami and a curved Miura origami when fully unfolded.** (a) a regular Miura origami ($\varphi_1 = \varphi_2$); (b) a curved Miura origami ($\varphi_1 \neq \varphi_2$).

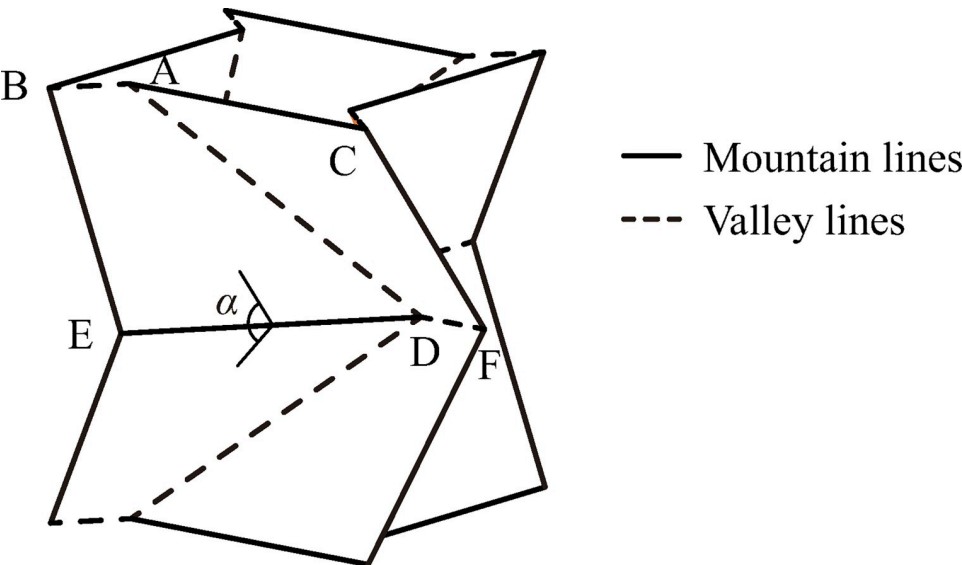

**Fig 2. A unit of a curved Miura origami-patterned tube.**

represent valley fold lines, $H$ represents the height of the pattern and $l_1$ and $l_2$ represent the respective side lengths ($l_1 \leq l_2$). In a regular Miura origami, the angle $\varphi_1$ remains equal to $\varphi_2$. Otherwise, the pattern is transformed to a curved Miura origami. The unequal $\varphi_1$ and $\varphi_2$ make the pattern globally curved and a curved Miura origami-patterned unit can be formed by connecting four pieces of the pattern end to end in order around an axis, once the closure condition given by Eq (1) [25, 26] is satisfied. The formed unit is shown in Fig 2, where $\alpha$ is the fold angle of the pattern.

$$\tan\varphi_1 \cdot \tan\varphi_2 \cdot \cos^2\frac{\alpha}{2} + \left(\tan\varphi_2 - \tan\varphi_1\right) \cdot \cos\frac{\alpha}{2} = -1 \qquad (1)$$

### 3-level curved Miura origami-patterned tubes

**Schemes of fold lines.** Rigid origami is an important concept about origami patterns, where the origami folds and unfolds with rigid panels rotating about hinge-like fold lines [27]. The hinge-like lines transfer axial force and shear but moment. Here we call them HL lines. However, in some cases, origami patterns are manufactured by cold forming techniques, including stamping against dies [19], cold gas pressure folding [28], and 3D printing [29]. In those manufacturing methods, materials are continuous at fold lines and the cross section area of materials there are not reduced. In such case, besides shear and axial force, the fold lines transfer moment. Thus, the lines can no longer be considered as frictionless hinges. Here those lines are named CT lines. By setting fold lines of the unit as one of the two types of lines we define above, four arrangement schemes of the lines are developed as shown in Fig 3.

**Numerical simulation validation.** Crash box and the tube studied in this work are both patterned by degree-4 vertices. A test on an origami crash box conducted in Zhou's work [30] is used to validate the finite element simulation method herein. In the work, a crash box was fixed on a rigid bearing plate and a parallel rigid loading plate moved vertically against the specimen at a rate of 0.83 mm/s, as shown in Fig 4. The specimen was made of 1 mm-thick steel plate at specified minimum yield stress of 235 MPa.

Universal finite element software package ANSYS/Implicit 18.0 is used to simulate the test process. Measured constitutive relationship of the steel is used for the simulation. Fig 5

| Scheme no. | Fold lines | | | Visualized scheme |
|---|---|---|---|---|
| | Horizontal lines | Inclined lines | | |
| | | Mountain lines | Valley lines | |
| 1 | HL | HL | HL | HL lines |
| 2 | CT | CT | CT | CT lines |
| 3 | HL | CT | CT | HL lines / CT lines |
| 4 | HL | HL | CT | HL lines / CT lines |

**Fig 3. Schemes of fold lines.**

presents the finite element modelling method. Element SHELL181, CONTA175 and TARGE170 are adopted and the boundary conditions are consistent with the test. Fig 6 compares the test results and the numerical results. It can be seen that the numerical results are in overall good agreement with the test results. The reason that the simulation exhibits greater initial stiffness is that the modelled crash box gets into perfect contact with the loading plate without any gap at the beginning of loading, while the specimen's end is imperfectly flat, leading to progressive contact. Severe strain concentration occurs at the vertex where four mountain lines and two valley lines meet in both results. The vertex is marked by a red dash square, where material fracture is most likely to occur. The agreement between the numerical results and the test results indicates the numerical simulation predicts the behavior of the specimen well.

**Nonlinear numerical simulations.** A curved Miura origami-patterned tube is assembled by vertically stacking three of the units shown in Fig 2. Four tubes are assembled corresponding to the four schemes listed in Fig 3, respectively. Since a given set of the parameters $H$, $l$ ($l_1+l_2$), $\varphi_1$ and $\alpha$ could determine the geometry of the pattern, identical parameters are adopted for the tubes, as listed in Table 1.

Nonlinear numerical simulations are conducted using ANSYS/Implicit 18.0, with the purpose to investigate buckling suppression of the schemes in Fig 3. The geometrical model and finite element modelling are detailed in Fig 7. Typical 4-node shell element SHELL181 is

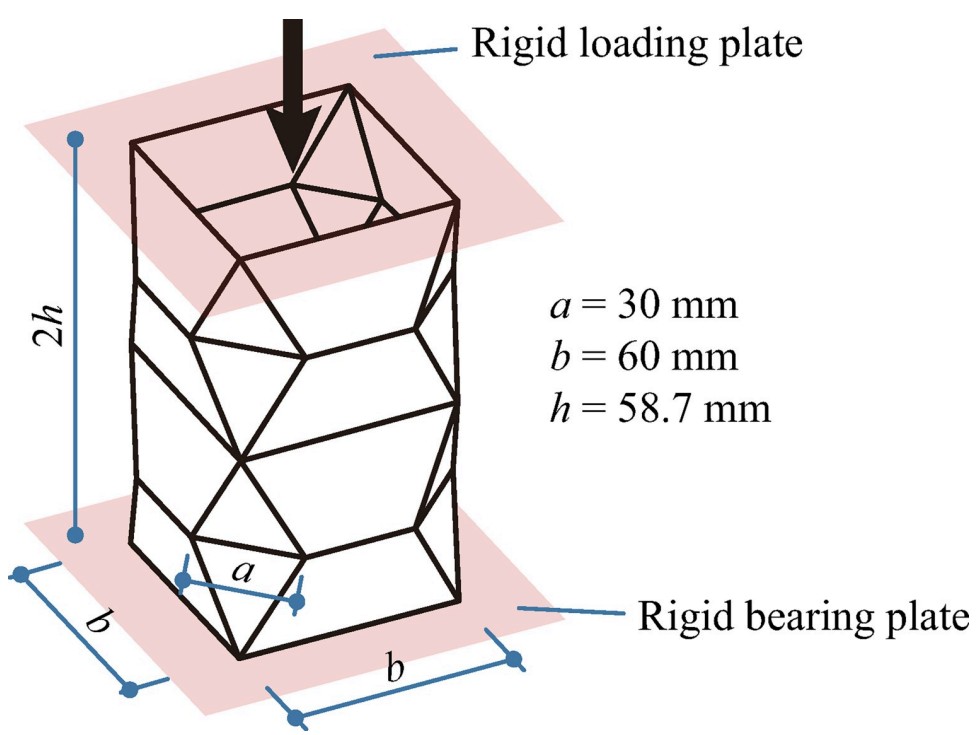

**Fig 4. Tested origami crash box.**

employed to simulate the 8 mm-thick panels of the origami pattern and the 20 mm-thick end plates of the tube. The end plates are used to close the hollow cross sections and increase local stability at ends. As a common practice for the connection between a brace and a beam-

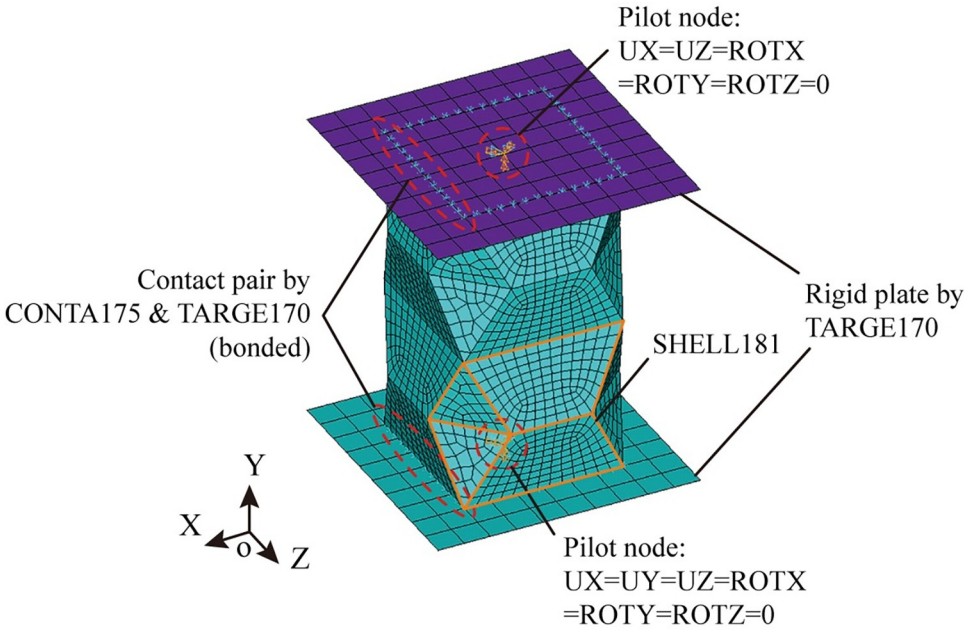

**Fig 5. Finite element modelling of crash box.**

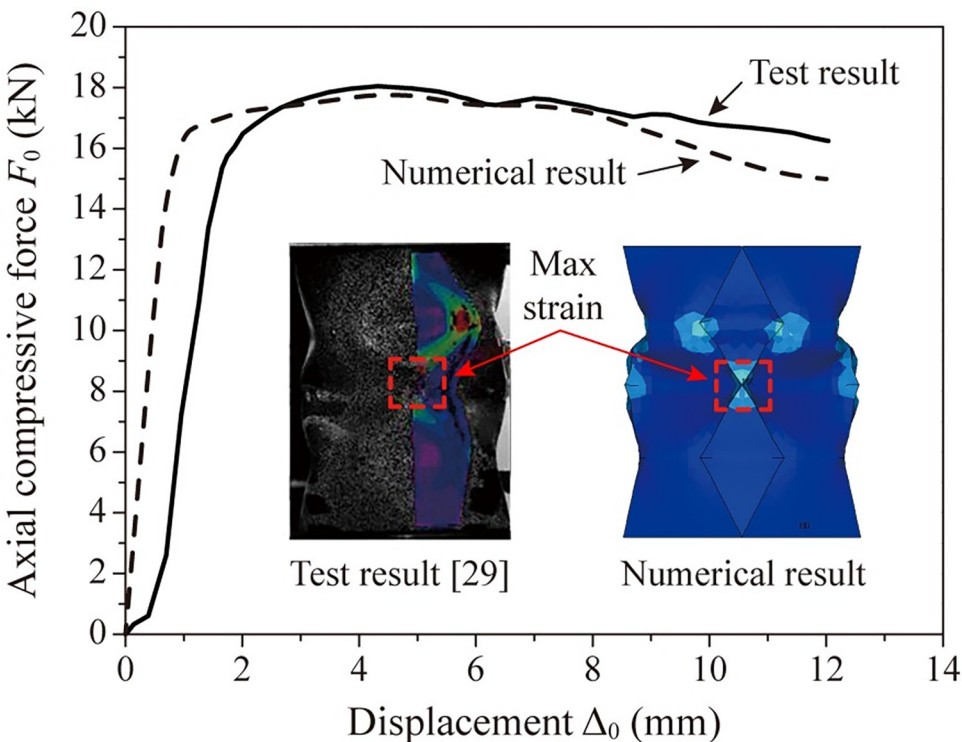

**Fig 6. Comparison between test results and numerical results.**

column joint in building structures, pin ends are used for the tubes. In order to simulate the pin ends, MASS21 element is placed at both ends, connecting nodes of the end plates by MPC184 Rigid Beam elements. In such way, with the constraints at MASS21 elements, rotations about the ends as a whole are allowed. The external load and the reaction force is applied at MASS21 elements. The loading end (the upper end) is allowed to translate in z direction and rotate about x and y directions (UX = UY = ROTZ = 0). The support (the lower end) is allowed to rotate about x and y directions (UX = UY = UZ = ROTZ = 0). For HL lines in the schemes, node coupling of the translational degrees of freedom along the lines are used. Thus, the lines are able to rotate without resistance. The material of the model is hot rolled steel with a 4-segment piecewise linear constitutive relationship [31], which is shown in Fig 8. The material parameters are listed in Table 2, where $f_y$ is the yield stress, $f_u$ is the ultimate stress, $E$ is the elastic modulus, $E_{st}$ is the slope of strengthening, $\varepsilon_y$ is the yield strain, $\varepsilon_{st}$ is the strain at the onset of stress strengthening, $\varepsilon_u$ is the strain at the ultimate stress and $\varepsilon_{el}$ is the ultimate strain. Displacement-controlled loading is applied in the simulation. In order to ensure the reach of a

**Table 1. Modelling information of patterned tubes.**

| Tube | Adopted scheme | Geometrical parameters | | | |
|---|---|---|---|---|---|
| | | $H$ / mm | $l\ (l_1+l_2)$ / mm | $\varphi_1$ / deg | $\alpha$ / deg |
| C-ORI-1 | Scheme 1 | 400 | 200 | 85 | 135 |
| C-ORI-2 | Scheme 2 | | | | |
| C-ORI-3 | Scheme 3 | | | | |
| C-ORI-4 | Scheme 4 | | | | |

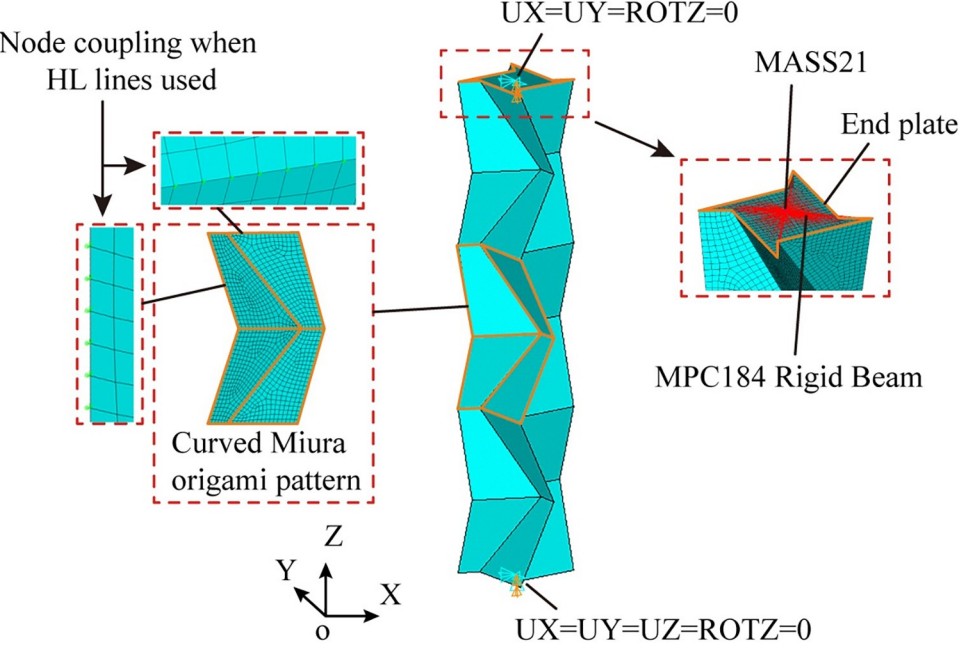

**Fig 7. Modelling of a 3-level patterned tube.**

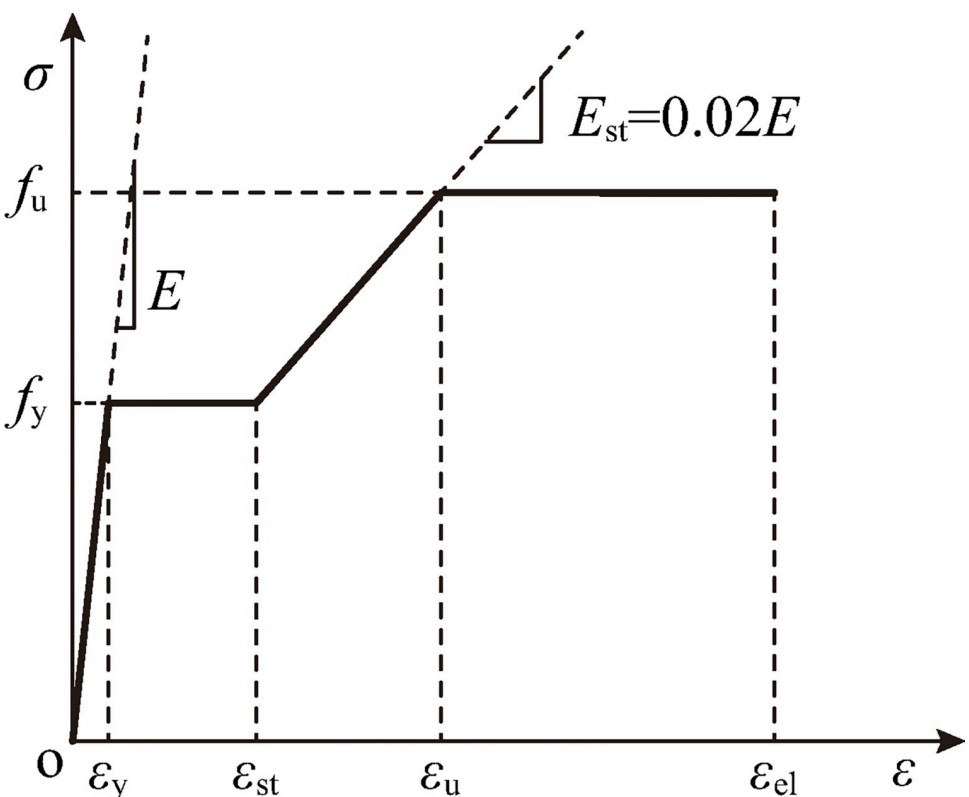

**Fig 8. Constitutive relationship of steel.**

**Table 2. Parameters of constitutive relationship of steel.**

| $E$ / MPa | $f_y$ / MPa | $f_u$ / MPa | $\varepsilon_y$ / % | $\varepsilon_{st}$ / % | $\varepsilon_u$ / % | $\varepsilon_{el}$ / % |
|---|---|---|---|---|---|---|
| 206000 | 260 | 420 | 0.13 | 1.51 | 5.39 | 26 |

peak point of the load force versus displacement curve, the loading distance is 60 mm (approximately 5.4% of the tube length).

## Results and discussion

The load force $F$ versus load displacement $\Delta$ curves of the tubes are shown in Fig 9, where the axial compressive strain $\varepsilon$ is calculated by Eq (2), the effective tube length $l_0$ = 1119 mm, and the peak points are symbolized. Deformation and VonMises stress distribution at the peak points and at the end of loading ($\Delta$ = 60 mm) are shown in Figs 10 and 11, respectively. In the figures, it can be seen that the tubes deform in a similar way and magnitude and distribution of the stress does not show much difference. The maximum stress appears at CT lines and the vertex near mid-length. For all the tubes, prior to peak points, just slight lateral deformation is observed and once the peak points are reached, the deformation increases rapidly. At $\Delta$ = 60 mm, global buckling of the tubes is significant. Figs 9–11 together demonstrate a discipline of the deformation transformation of the tubes: before the peak points, the deformation modes are governed by axial compression with mild global bending; after the points are reached, a

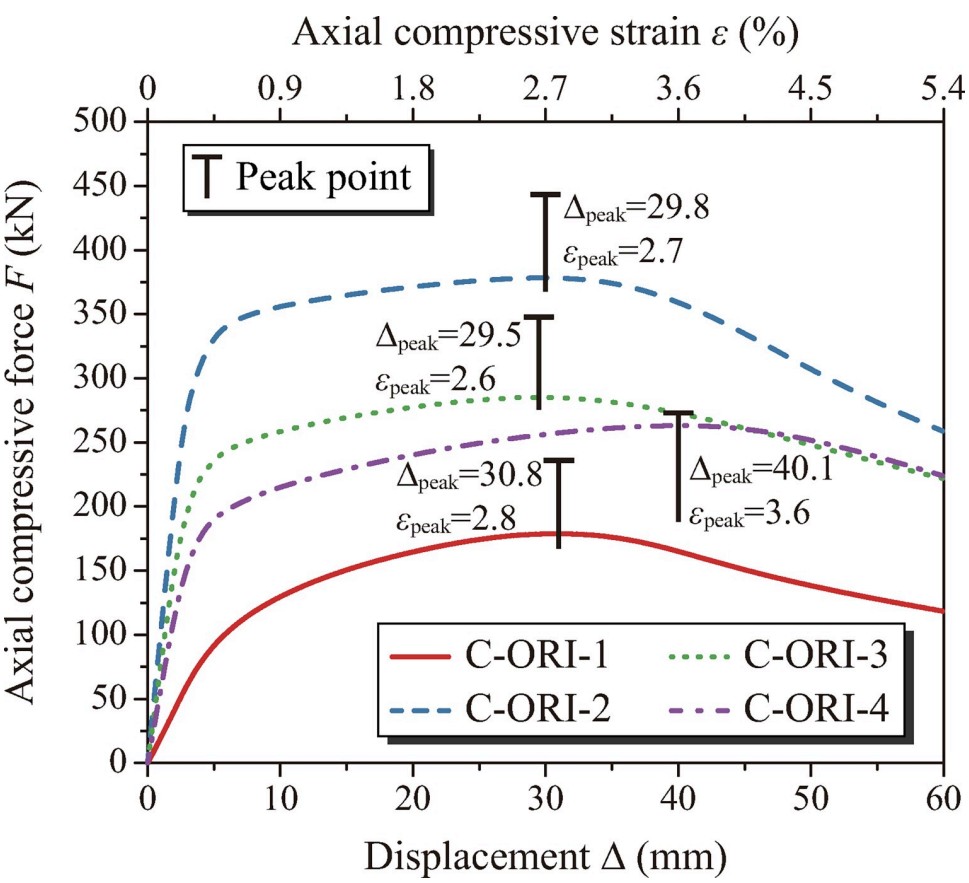

**Fig 9. Load force $F$ versus load displacement $\Delta$ curves.**

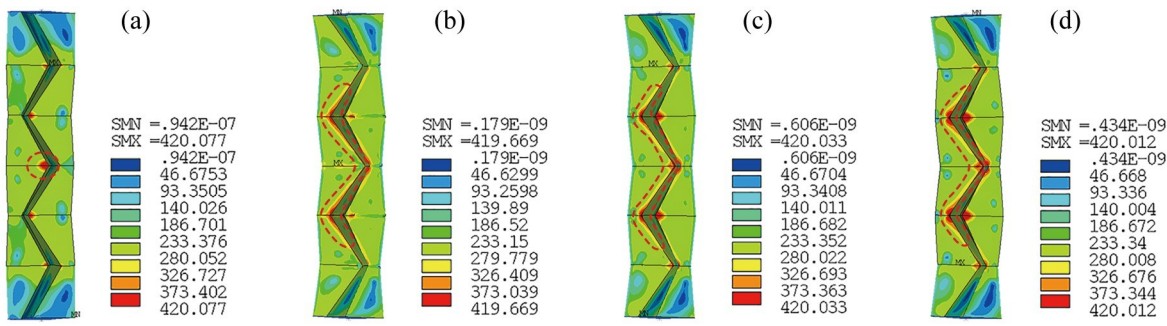

**Fig 10. Deformation and stress distribution of tubes at peak points.** (a) C-ORI-1; (b) C-ORI-2; (c) C-ORI-3; (d) C-ORI-4.

swift in-crease of the lateral deformation is experienced, and the major deformation mode transfers to global flexural buckling, accompanied by substantial capacity loss and consequently the tubes' instability. Values of displacement at the peak point $\Delta_{peak}$ and the corresponding axial compressive strain $\varepsilon_{peak}$ are given next to the peak points. It can be found that the values of $\Delta_{peak}$ and $\varepsilon_{peak}$ of C-ORI-4 are 30.2%, 34.6% and 35.9% larger than those of C-ORI-1, C-ORI-2 and C-ORI-3 respectively, indicating the greatest axial deformation capacity of C-ORI-4 among the tubes.

$$\varepsilon = \frac{\Delta}{l_0} \qquad (2)$$

Consistent with typical simply-supported axially loaded column, the global bending of the patterned tubes is found to be in a semi-sine wave. In order to indicate the magnitude of deviation from the initial straight states, curvature of the deformed tube $k$ is adopted as Eq (3), where $\theta_1$ and $\theta_2$ are the rotations of the support end and the loading end respectively, as shown in Fig 12(A). Fig 12(B) depicts the values of $k$ during the load process. It can be seen in the figure that for all the tubes the values of $k$ remain small and go up almost linearly as the displacement $\Delta$ increases before peak points are arrived, while the increase is much accelerated in the post peak periods. C-ORI-4 is observed to behave the least $k$ among the tubes. Specifically, the maximum rotation of C-ORI-4 are 6.5 degrees and for any given displacement $\Delta$ in the load process, the curvature of C-ORI-4 are up to 59.9%, 57.3% and 52.3% less than those of C-ORI-1, C-ORI-2 and C-ORI-3 respectively. The least deviation from the initial straight states indicates the least magnitude of lateral deformation and global buckling of C-ORI-4 among the tubes in the post peak periods. Besides, it is the least curvature that leads to the

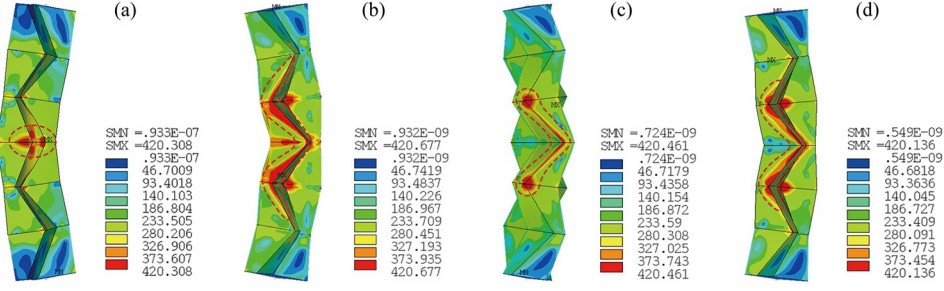

**Fig 11. Deformation and stress distribution of tubes at $\Delta = 60$ mm.** (a) C-ORI-1; (b) C-ORI-2; (c) C-ORI-3; (d) C-ORI-4.

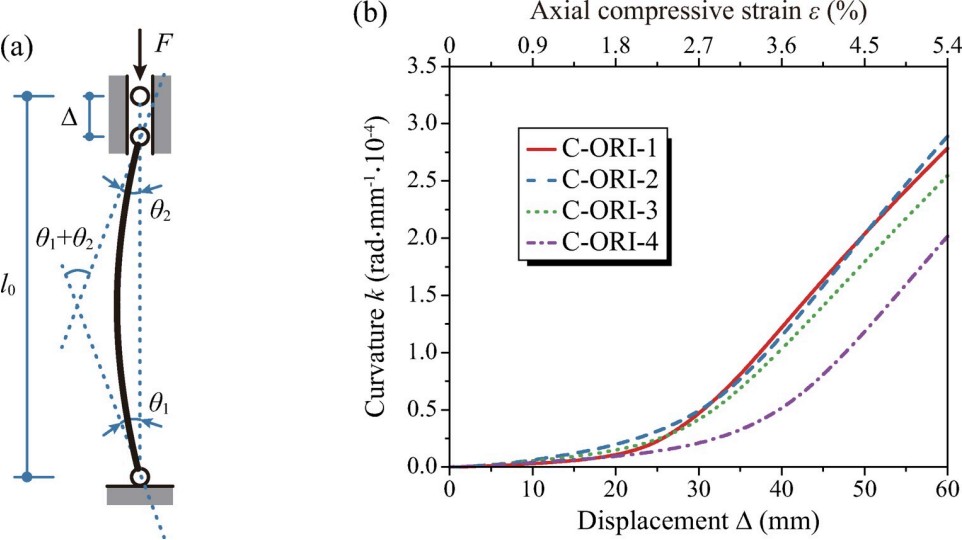

**Fig 12. Rotations at ends.** (a) illustration of rotations; (b) curvatures during loading process.

latest occurrence of global buckling of C-ORI-4 as shown in Fig 9.

$$k = \frac{\theta_1 + \theta_2}{l_0} \tag{3}$$

Through the observations and discussions above, it is demonstrated that global buckling of C-ORI-4 is suppressed to most extent among the four tubes investigated. Therefore, in light of the same geometry shared by the tubes, Scheme 4, represented by C-ORI-4, is believed to be an effective strategy to suppress global buckling of a curved Miura origami-patterned tube.

## Application to a long tube

To extend the work to design of braces in building structures, we apply Scheme 4 to a long tube in this section. Eight units are vertically stacked, increasing the tube length to $L_0 = 3400$ mm. Distinct geometrical configurations are adopted and the parameters are listed in Table 3. The thickness of the steel panel is 8 mm and that of the end plate is 20 mm. The same modelling method is adopted as previously mentioned and is shown in Fig 13.

Numerical simulation is conducted on the mechanical behavior of the tubes under axial compression. Table 4 gives the values of $\Delta_{peak}$ and $\varepsilon_{peak}$, which indicate occurrence of global

**Table 3. Modelling information of patterned tubes.**

| Tube | Geometrical parameters | | | |
|---|---|---|---|---|
| | $H$ / mm | $l$ $(l_1+l_2)$ / mm | $\varphi_1$ / deg | $\alpha$ / deg |
| C-ORI-L-1 | 200 | 280 | 80 | 120 |
| C-ORI-L-2 | 200 | 300 | 80 | 120 |
| C-ORI-L-3 | 200 | 280 | 85 | 135 |
| C-ORI-L-4 | 220 | 300 | 80 | 120 |
| C-ORI-L-5 | 220 | 300 | 80 | 135 |
| C-ORI-L-6 | 240 | 280 | 85 | 105 |

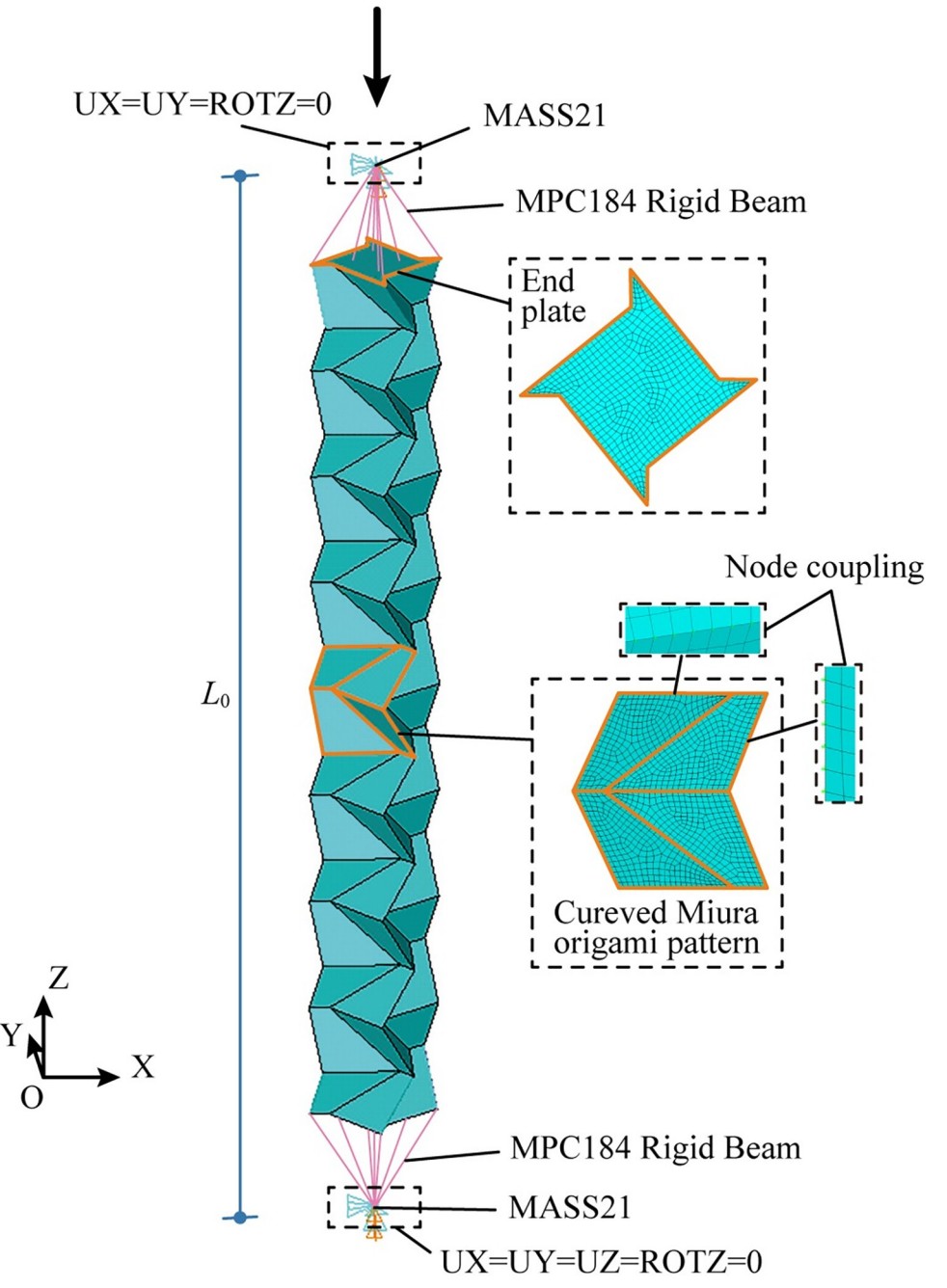

**Fig 13. Modelling of an 8-level patterned tube.**

buckling. It can be seen in the table that the values of C-ORI-L-1, C-ORI-L-2 and C-ORI-L-4 are substantially larger than the others (up to 6 times as those of the others), indicating that global buckling occurs much later for the three tubes. Deformation and stress distribution of the three tubes at $\Delta_{\text{peak}}$ and $\varepsilon_{\text{peak}}$ are presented in Fig 14. It can be observed that the tubes deform axially and hardly bent, with uniformly stress distribution in length at that large axial compressive strain, which is barely found in conventional tubes in civil engineering (straight square and circular long tubes). It demonstrates the superior axial deformation capacity of the

**Table 4. Values of $\Delta_{peak}$ and $\varepsilon_{peak}$.**

| Tube | $\Delta_{peak}$ / mm | $\varepsilon_{peak}$ / % |
|---|---|---|
| C-ORI-L-1 | 80.96 | 2.38 |
| C-ORI-L-2 | 90.04 | 2.65 |
| C-ORI-L-3 | 27.61 | 0.81 |
| C-ORI-L-4 | 93.02 | 2.74 |
| C-ORI-L-5 | 15.61 | 0.46 |
| C-ORI-L-6 | 23.61 | 0.69 |

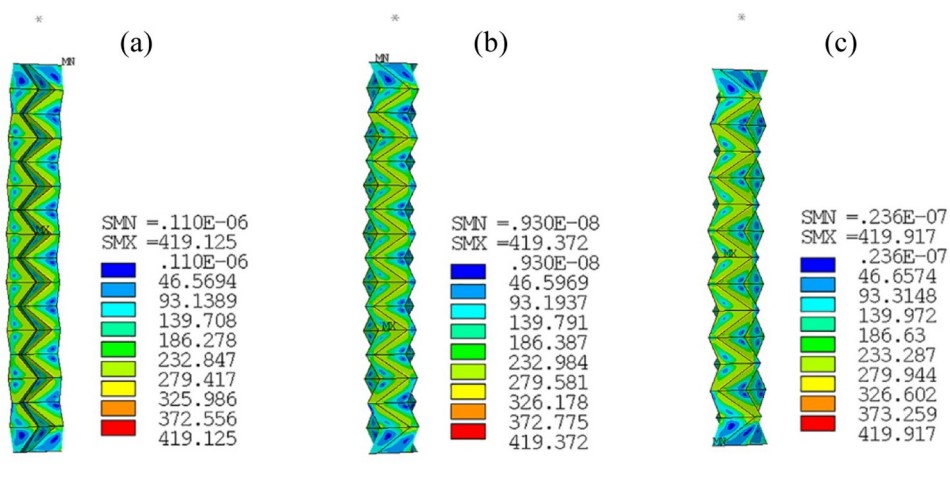

**Fig 14. Deformation and stress distribution of tubes at peak points.** (a) C-ORI-L-1; (b) C-ORI-L-2; (c) C-ORI-L-4.

tubes to the conventional tubes and proves that Scheme 4 is potentially an effective way to alleviate buckling instability of a long tube when appropriately designed.

## Conclusions

Global buckling suppression is an important issue for energy absorbing tubes, as the buckling alters the deformation mechanism and in turn deteriorates the energy absorption capacity. The present work provides a novel strategy to effectively suppress global buckling of a curved Miura origami-patterned tube by applying a specific scheme of fold lines of the pattern. According to the simulation results, the scheme manages to reduce the lateral deformation in roughly half compared with the counterparts. Conclusions from the work are drawn as follows:

- By categorizing the fold lines of curved Miura origami into HL lines and CT lines, different schemes of the octagon patterned tube can be developed. Tubes using the schemes behave distinct lateral deformations under axial compression. By taking advantage of this feature, a superior scheme can be found to achieve the least lateral deformation and the best axial deformation capacity.

- Compared with the other fold line schemes, it is found that when the inclined valley lines are considered as HL lines and the others as CT lines, the scheme enables the 3-level tube to reduce the magnitude of lateral deformation by up to 59.9% and to delay the occurrence of

global buckling by up to 35.9%. It indicates the excellence of the scheme in global buckling suppression of the patterned tube.

- Provided appropriate design, a long patterned tube using the superior scheme can achieve excellent axial deformation capacity under axial compression with substantially delayed global buckling, unnoticeable lateral deformation and uniformly stress distribution in length. The scheme is potentially an effective way to alleviate buckling instability of a long tube.

In follow-up study, the scheme will be parametrically examined on slender tubes and preparation of a physical model will be on schedule.

## Supporting information

**S1 File. Original images of Figs 6, 10–12.**
(DOCX)

**S1 Table. Data behind Fig 6.**
(XLSX)

**S2 Table. Data behind Fig 9.**
(XLSX)

**S3 Table. Data behind Fig 12.**
(XLS)

**S4 Table. Data behind Table 4.**
(XLSX)

## Acknowledgments

Special thanks to Yang Li from Wuhan University for his suggestions about this work. Administrative and technical supports from Chongqing Jiaotong University and Southeast University are gratefully acknowledged.

## Author Contributions

**Conceptualization:** Yangqing Liu, Jianguo Cai.

**Data curation:** Yangqing Liu.

**Formal analysis:** Yangqing Liu.

**Funding acquisition:** Yangqing Liu, Jianguo Cai, Jian Feng.

**Investigation:** Yangqing Liu.

**Methodology:** Yangqing Liu.

**Project administration:** Yangqing Liu, Jianguo Cai, Jian Feng.

**Resources:** Yangqing Liu.

**Software:** Yangqing Liu.

**Supervision:** Jianguo Cai, Jian Feng.

**Validation:** Yangqing Liu, Jianguo Cai, Jian Feng.

**Visualization:** Yangqing Liu.

Writing – **original draft:** Yangqing Liu.

Writing – **review & editing:** Yangqing Liu.

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
