## [Decision Letter · Decision Letter 0]

9 Mar 2022

PONE-D-21-23372Buckling suppression of a thin-walled Miura-origami patterned tubePLOS ONE

Dear Dr. Liu,

Thank you for submitting your manuscript to PLOS ONE. After careful consideration, we feel that it has merit but does not fully meet PLOS ONE’s publication criteria as it currently stands. Therefore, we invite you to submit a revised version of the manuscript that addresses the points raised during the review process.

We look forward to receiving your revised manuscript.

Kind regards,

Jianguo Wang, PhD

Academic Editor

PLOS ONE

Journal Requirements:

Reviewers' comments:

Reviewer's Responses to Questions

**Comments to the Author**

1. Is the manuscript technically sound, and do the data support the conclusions?

Reviewer #1: Partly

Reviewer #2: Yes

2. Has the statistical analysis been performed appropriately and rigorously? 

Reviewer #1: N/A

Reviewer #2: N/A

3. Have the authors made all data underlying the findings in their manuscript fully available?

Reviewer #1: No

Reviewer #2: Yes

4. Is the manuscript presented in an intelligible fashion and written in standard English?

Reviewer #1: Yes

Reviewer #2: Yes

5. Review Comments to the Author

Reviewer #1: This paper presents a novel fold-line scheme for delaying the onset of global buckling in long

Curved Miura tubes. The Curved Miura tubes have an octagonal cross-section and the material

properties of steel. Each mountain and valley crease of these tubes is categorized as either a

hinge-like (HL) or a continuous (CT) fold line. HL lines only transfer the axial and shear forces

in the tube but they do not transfer moments along the crease. Continuous lines maintain the

material properties and thickness and of the tube and can transfer moments in addition to the

axial and shear forces. Four schemes are investigated in this paper: 1. All fold-lines are HL, 2.

All fold-lines are CT, 3. Only horizontal fold-lines are HL, and 4. Only slanted valley creases are

CT. In summary, the authors’ numerical model successfully showed that the different schemes

affect when global buckling is initiated in the tubes and that the fourth scheme appears to provide

the greatest suppression of the global buckling. The reviewer saw the novelty of the work for

Curved Miura tubes. Also, the manuscript is well organized. However, there are a few areas

where clarification and additional justification for the research approach taken is necessary to

fully understand the work. The authors should try to address the following points before the

paper is published:

It is not clear why the authors’ selected the Curved Miura architecture for study. Miura

fold patterns are usually utilized for applications concerning deployability due to their

rigid-foldable behavior which, usually does not make them suitable for energy absorption

applications. Other geometries such as Yoshimura, Kresling, variations of square patterns

(including one used for Numerical Verification in this paper) and others have been

investigated by other groups in the past because these geometries can take advantage of

deformations in the facets as well as the creases for energy absorption.

Energy absorption tubes usually do not use hinges or other compliant mechanisms. Even

if global buckling can be suppressed through all of compression, can one achieve enough

energy absorption to make Curved Miura with Scheme 4 competitive with other tube

geometries?

Are Schemes 1-4 the only ones that can be generated? Could a Scheme 5 include CT

horizontal creases but all other fold line are HL? Are the schemes selected based on their

possible manufacturability?

It is skeptical that the test conducted on the crash box from Zhou’s work counts as

numerical validation for this paper. Certain boundary conditions are not compatible with

the Curved Miura tube. Additionally, the Curved Miura tube simulations require node

couplings along creases for Schemes 1, 3, and 4 and all simulations use beams at the ends

of the tubes. Plus, for reasons not quite stated, the ends of the tubes have a greater

thickness than the walls of the tube. Curved Miura is also an entirely different geometry.

More justification should be given for this step.

A comment on the choice for the height of the tubes would also be helpful. Although

there is a section of the paper entitled “Application to a Long Tube,” the tubes before this

section are already significantly longer than other pre-folded tubes in many energy

absorption studies. If the tubes with H=400 mm can trigger global buckling and are

already longer than most tubes studied, why is there another section devoted to tubes that

are even longer?

Other comments the authors may find helpful:

In the Abstract, there is a sentence that says “delays global buckling by ~36%” but this

number has no context. Is it ~36% compared to some control tube? This also happens in

the Conclusion when it is said that Scheme 4 reduces the lateral deformation “in half.” In

half compared to what?

“Unusual” is a strong word to use and is not elaborated upon in the Introduction. What

properties are “unusual”?

“Except that” does not work for its respective sentence in the Introduction. Maybe “Also”

could be used instead?

“Highly sensitivity” was probably meant to be “High sensitivity”

In the Introduction, the work done by Guest and Pellegrino is used as an example of the

Kresling pattern being applied to thin-walled tubes but the work was not conducted for

energy absorption applications. It may be beneficial to add a reference for the research

completed by Professor Ichiro Hagiwara on metal Kresling tubes for energy absorption

applications.

In the Materials and Methods section, there is one part that reads “…transfer axial force

and shear but moment.” I think this part is supposed to say “…but not moment.”

In Figure 3, why are the horizontal lines at the top and bottom cross-sections categorized?

The reviewer’s understanding is that these are not creases.

Why are facets 8mm thick in numerical simulations? Also, what are the dimensions of

l_1 and l_2?

In the Results and Discussion, there is one line that reads “…as shown in Fig. 8” but this

figure does not include data. Is this a typo?

In the Application to a Long Tube section, “conventional tubes” is used as a term but

how is this defined? Are these straight-walled tubes? Shorter tubes? Other pre-folded

tubes?

Reviewer #2: The work proposes an origami folding pattern to supress global buckling of a thin-walled tube. It is a very interesting and effective concept. There are some minor issues that need to be clarified.

1. "curved Miura pattern" is used. I wonder if here "curved" may be misunderstood as "curved" creases. If this is not a common term, the authors may need to think of another term.

2. In the finite element modelling of a crash box, details of the material properties should be given, in addition to the value of yield stress. Was the actual sress-strain curve used, or was a bi-linear model used, for example.

3. it would be interesting to discuss about tubes of the same/similar cross-sectional dimensions when the effect of patterning was studied. This needs to be mentioned.

4. It was not clear if global buckling was actually supressed or has just been delayed. It would be interesting to see how effectvie the method is for tubes of different lengths.

6. PLOS authors have the option to publish the peer review history of their article (what does this mean?). If published, this will include your full peer review and any attached files.

Reviewer #1: No

Reviewer #2: No

---

## [Author Response · Author response to Decision Letter 0]

22 Apr 2022

Dear Editor and Reviewers,

Thank you for your letter and for the reviewers’ comments concerning our manuscript entitled “Buckling suppression of a thin-walled Miura-origami patterned tube” (ID: PONE-D-21-23372). The comments are valuable and constructive for the improvement of the manuscript, and of significant importance to our research work.

We have revised our manuscript after reading the comments from the reviewers. The revisions include content revisions and language improvement. The revised portions are tracked in the uploaded file labeled ‘Revised Manuscript with Track Changes’. An unmarked version of the revised manuscript labeled ‘Manuscript’ is also uploaded.

The response to the reviewers is uploaded labeled as ‘Response to Reviewers’, which includes figures for explanations.

Lastly, the study’s minimal underlying data set is uploaded as a Supporting Information file.

Best,

Dr. Yangqing Liu & Dr. Jianguo Cai

School of Civil Engineering

Chongqing Jiatong University & Southeast University

Chongqing, P.R.China & Nanjing, Jiangsu, P.R.China

---

## [Decision Letter · Decision Letter 1]

7 Jun 2022

Buckling suppression of a thin-walled Miura-origami patterned tube

PONE-D-21-23372R1

Dear Dr. Liu,

We’re pleased to inform you that your manuscript has been judged scientifically suitable for publication and will be formally accepted for publication once it meets all outstanding technical requirements.

Kind regards,

Jianguo Wang, PhD

Academic Editor

PLOS ONE

Additional Editor Comments (optional):

Reviewers' comments:

Reviewer's Responses to Questions

**Comments to the Author**

1. If the authors have adequately addressed your comments raised in a previous round of review and you feel that this manuscript is now acceptable for publication, you may indicate that here to bypass the “Comments to the Author” section, enter your conflict of interest statement in the “Confidential to Editor” section, and submit your "Accept" recommendation.

Reviewer #1: All comments have been addressed

Reviewer #2: All comments have been addressed

2. Is the manuscript technically sound, and do the data support the conclusions?

Reviewer #1: Yes

Reviewer #2: Yes

3. Has the statistical analysis been performed appropriately and rigorously? 

Reviewer #1: Yes

Reviewer #2: N/A

4. Have the authors made all data underlying the findings in their manuscript fully available?

Reviewer #1: Yes

Reviewer #2: Yes

5. Is the manuscript presented in an intelligible fashion and written in standard English?

Reviewer #1: Yes

Reviewer #2: Yes

6. Review Comments to the Author

Reviewer #1: (No Response)

Reviewer #2: the authors have addressed my points. the manuscript has been revised accordinly, although it would have been better to incorporate the points in the revision in addition to response part.

7. PLOS authors have the option to publish the peer review history of their article (what does this mean?). If published, this will include your full peer review and any attached files.

Reviewer #1: No

Reviewer #2: No

---

## [Editor Report · Acceptance letter]

10 Jun 2022

PONE-D-21-23372R1 

Buckling suppression of a thin-walled Miura-origami patterned tube 

Dear Dr. Liu:

I'm pleased to inform you that your manuscript has been deemed suitable for publication in PLOS ONE. Congratulations! Your manuscript is now with our production department. 

Kind regards, 

on behalf of

Dr. Jianguo Wang 

Academic Editor

PLOS ONE